# PRIVATE SPLIT INFERENCE OF DEEP NETWORKS

## ABSTRACT

Splitting network computations between the edge device and the cloud server is a promising approach for enabling low edge-compute and private inference of neural networks. Current methods for providing the privacy train the model to minimize information leakage for a given set of private attributes. In practice, however, the test queries might contain private attributes that are not foreseen during training. We propose an alternative solution, in which, instead of obfuscating the information corresponding to a set of attributes, the edge device discards the information irrelevant to the main task. To this end, the edge device runs the model up to a split layer determined based on its computational capacity and then removes the activation content that is in the null space of the next layer of the model before sending it to the server. It can further remove the low-energy components of the remaining signal to improve the privacy at the cost of reducing the accuracy. The experimental results show that our methods provide privacy while maintaining the accuracy and introducing only a small computational overhead.

## 1 INTRODUCTION

The surge in cloud computing and machine learning in recent years has led to the emergence of Machine Learning as a Service (MLaaS), where the compute capacity of the cloud is used to analyze the data that lives on edge devices. One shortcoming of the MLaaS framework is the leakage of the clients' private data to the cloud server. To address this problem, several cryptography-based solutions have been proposed which provide provable security at the cost of increasing the communication cost and delay of remote inference by orders of magnitude (Juvekar et al. (2018); Riazi et al. (2019)). The cryptography-based solutions are applicable in use-cases such as healthcare where a few minutes of delay is tolerable, but not in scenarios where millions of clients request fast and low-cost responses such as in Amazon Alexa or Apple Siri applications. A light-weight alternative to cryptographic solutions is to manually hide private information on the edge device; For instance, sensitive information in an image can be blurred on the edge device before sending it to the service provider (Vishwamitra et al. (2017)). This approach, however, is task-specific and may not be viable for generic applications.

The objective of split inference framework, shown in Figure 1, is to provide a generic and computationally efficient data obfuscation scheme (Kang et al. (2017); Chi et al. (2018)). The service provider trains the model and splits it into two sub-models, $M_1$ and $M_2$, where $M_1$ contains the first few layers of the model and $M_2$ contains the rest. The client runs $M_1$ on the edge device and sends the resulting feature vector $z = M_1(x)$ to the server, which computes the public label as $y^{\mathrm{pub}} = M_2(z)$. To preserve the privacy, the client desires $z$ to only contain information related to the underlying task. For instance, when sending facial features for cell-phone authentication, the client does not want to disclose other information such as their mood. As seen in Figure 1, the privacy leakage is quantified by an adversary that trains the model $M_3$ to extract private label $y^{\mathrm{pri}}$ from feature vector $z$.

Current methods of private split inference aim to censor the information corresponding to a list of known private attributes. For example, Feutry et al. (2018) utilize adversarial training to minimize the accuracy of $M_3$ on the private attribute, and Osia et al. (2018) minimize the mutual information between the query $z$ and the private label $y^{\mathrm{pri}}$ at training time. The set of private attributes, however, can vary from one query to another. Hence, it is not feasible to foresee all types of attributes that could be considered private for a specific MLaaS application. Moreover, the need to annotate inputs with all possible private attributes significantly increases the cost of model training.

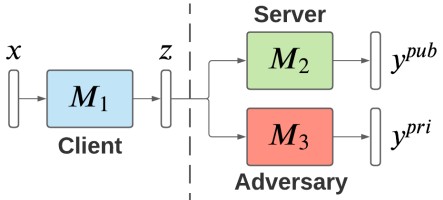

Figure 1: Split inference setup. Client runs $M_1$ locally and sends the features $z = M_1(x)$ to the server. The server predicts the intended attribute as $y^{\mathrm{pub}} = M_2(z)$. An adversary trains a separate model $M_3$ to predict the private attribute as $y^{\mathrm{pri}} = M_3(z)$.

In this paper, we propose an alternative solution where, instead of censoring the information that is utilized to predict known private attributes, we discard the information that is not used by the main model for predicting the public attribute. Our contributions are summarized in the following.

- We characterize the information that is not relevant to the prediction of the public attribute as part of the content of the feature vector $z$ that will be removed by the server-side model. We then define the null content of the feature vector, $z_{\mathcal{N}}$, as the content in $z$ that is in the null-space of the following linear layer. The remaining content is called signal content and is denoted by $z_{\mathcal{S}}$. We have $M_2(z) = M_2(z_{\mathcal{S}} + z_{\mathcal{N}}) = M_2(z_{\mathcal{S}})$.

- We propose to remove $z_{\mathcal{N}}$ from features, $z$, and show that it reduces the accuracy of the adversary ($M_3$), while maintaining the accuracy of the main model ($M_2$). To further discard the private information in $z$, we propose to remove the low-energy components of $z_{\mathcal{S}}$, through which we achieve higher privacy (lower accuracy of $M_3$) at the cost of a small reduction in utility (lower accuracy of $M_2$).

- We show our methods provide tradeoffs between edge-computation efficiency, privacy, and accuracy. Specifically, with higher edge computation (more layers on the edge device), the client achieves better privacy at the same accuracy. Also, with the same edge computation (a given split layer), removing more components from the signal content provides better privacy at the cost of reduced accuracy.

- We perform extensive experiments on several datasets and show that our methods provide better tradeoffs between accuracy and privacy compared to existing approaches such as adversarial training, despite having no knowledge of the private attribute at training or inference times.

## 2 BACKGROUND AND RELATED WORK

We consider the supervised learning setting of Figure 1, where the model $M_2 \circ M_1$ is trained with a set of examples $\{x_i\}_{i=1}^N$ and their corresponding public labels $\{y_i^{\mathrm{pub}}\}_{i=1}^N$. At inference phase, the client runs $M_1$ on their data and sends the intermediate feature vector $z = M_1(x)$ to the server. The goal of private inference is to ensure that $z$ does not contain information about private attributes.

### 2.1 MEASURING PRIVACY

Several methods have been proposed to measure the privacy leakage of the feature vector. One approach is computing the mutual information between the query $x$ and the feature vector $z$ (Kraskov et al. (2004)). In practice, measuring the mutual information is not tractable for high-dimensional random variables, unless certain assumptions are made about the probability distribution of the random variables of interest. A more practical approach measures privacy based on the reconstruction error, $\|\widetilde{x} - x\|$, where $\widetilde{x}$ is estimated based on $z$ (Mahendran & Vedaldi (2015)). Finally, attribute privacy is defined based on the accuracy of an adversary model that takes $z$ as input and predicts the private label.

In this paper, we use the attribute privacy notion as it applies to a wide range of applications. Assume each example $\{x_i\}_{i=1}^N$ has one or multiple private labels $\{y_i^{\mathrm{pri}}\}_{i=1}^N$. The adversary trains a separate model $M_3$ with $(z_i, y_i^{\mathrm{pri}})$ where $z_i = M_1(x_i)$, as shown in Figure 1. Note that $M_3$ is used as an aftermath process to evaluate the privacy of the model $M_2 \circ M_1$. The split learning framework should achieve high *utility*, i.e., the server should be able to infer the public attribute from $z$ accurately, while providing *privacy*, i.e., $z$ should not contain information about $y^{\mathrm{pri}}$. We refer to the accuracy of $M_2 \circ M_1$ on $y^{\mathrm{pub}}$ as public accuracy and the accuracy of $M_3 \circ M_1$ on $y^{\mathrm{pri}}$ as private accuracy.

## 2.2 THREAT MODEL

**Honest-but-curious server.** The server performs the inference of the public attribute but will potentially try to extract private information from the features, $z$, as well.

**Client capabilities.** Upon providing the service, the server also provides a profile of the utility (accuracy on the public attributes), privacy (accuracy on several private attributes), and computation of the edge device. The client then decides on the best tradeoff based on the computational resources and also the desired level of privacy. Such mechanisms are already in use in ML-on-the-edge applications. For example, in the application of unlocking the phone by face recognition, the client can specify the required precision in face recognition, where a lower precision will provide higher utility at the cost of lower security (Chokkattu, 2019).

## 2.3 RELATED WORK

Prior work has shown that the representations learned by deep neural networks can be used to extract private information (Song & Shmatikov (2019)) or even reconstruct the raw data (Mahendran & Vedaldi (2015)). Current methods for private inference can be categorized as follows.

**Cryptography-based solutions.** Since the server is not trusted, solutions based on public key encryption (Al-Riyami & Paterson (2003)) are not applicable. We consider scenarios where the server provides service to millions of users (e.g., in cases of Amazon Alexa or Apple Siri), and users expect low communication and fast response. Therefore, classic two-party cryptographic solutions for secure function evaluation (Juvekar et al. (2018); Riazi et al. (2019)) are also not applicable to our scenario.

**Noise Injection.** A line of work suggests obfuscating private attributes by adding noise to the features, i.e., instead of $z$, the client sends $z + \mu$ to the server, with the noise designed to maintain public accuracy while reducing private accuracy (Mireshghallah et al. (2020)). While noise addition improves privacy, it has been shown to reduce the public accuracy significantly (Liu et al. (2019)).

**Information Bottleneck.** The notion of mutual information can be used to train private models. Let $I(a, b)$ denote the mutual information between random variables $a$ and $b$. The idea is to train $M_1$ that maximizes $I(z, y^{\mathrm{pub}})$ while minimizing $I(z, y^{\mathrm{pri}})$ (Osia et al. (2018); Moyer et al. (2018)). The optimization is formulated as follows:

$$\max_{M_1} \quad \mathbb{E}_{x, y^{\mathrm{pub}}, y^{\mathrm{pri}}}[I(M_1(x), y^{\mathrm{pub}}) - \gamma I(M_1(x), y^{\mathrm{pri}}) - \beta I(M_1(x), x)]. \tag{1}$$

The use of mutual information for privacy, however, has been challenged by practical attacks that extract secret information even when $I(z, y^{\mathrm{pri}})$ is small (Song & Shmatikov (2019)).

**Adversarial Training.** This defense solves the following min-max optimization problem:

$$\max_{M_1, M_2} \min_{M_3} \mathbb{E}_{x, y^{\mathrm{pub}}, y^{\mathrm{pri}}}[\gamma \mathcal{L}(y^{\mathrm{pri}}, M_3 \circ M_1(x)) - \mathcal{L}(y^{\mathrm{pub}}, M_2 \circ M_1(x))], \tag{2}$$

where $\mathcal{L}$ denotes the cross-entropy loss and $\gamma$ is a scalar. The above objective can be achieved through adversarial training (Edwards & Storkey (2016); Hamm (2017); Xie et al. (2017); Li et al. (2018); Feutry et al. (2018); Li et al. (2019)). At convergence, the trained $M_1$ generates $z$ such that $M_3(z)$ is not an accurate estimation of $y^{\mathrm{pri}}$ while $M_2(z)$ accurately describes $y^{\mathrm{pub}}$.

Existing methods for private split inference have several limitations. First, the underlying assumption in above learning-based defenses is that a set of private attributes along with the public label are provided at training time. In practice, however, it might not be feasible to foresee and identify all possible private attributes and annotate the training data accordingly. It also contradicts deployment at-scale since whenever a new private attribute is identified, the model $M_1$ needs to be retrained and re-distributed to all edge devices that use the service. Second, current approaches for private inference often provide a poor tradeoff between accuracy and privacy. Moreover, the tradeoff of accuracy and privacy with the client-side computation is not well studied in the split learning framework. In this paper, we characterize this tradeoff and propose an alternative approach that, instead of obfuscating the information related to the private attributes, the edge device removes the feature content that is irrelevant to the public task. We empirically show our method successfully reduces the accuracy on private attributes at a small or no cost to public accuracy.

## 3 PROPOSED METHODS

### 3.1 SIGNAL AND NULL CONTENTS OF FEATURE VECTOR

Let $z \in \mathbb{R}^n$ be a feature vector and $W \in \mathbb{R}^{m \times n}$ be a matrix. The operation of fully-connected and convolutionals layers can be expressed as matrix-vector and matrix-matrix multiplication. Herein, we let $z$ represent the vector in fully-connected layer and a column of the second matrix in convolution layer. Let the singular value decomposition (SVD) of $W$ be $W = U \cdot S \cdot V$. Since the rows of $V$ form an orthonormal basis, we can write the feature vector as

$$z = \sum_{i=1}^{n} \alpha_i v_i^T, \qquad \alpha_i = < v_i^T, z >, \tag{3}$$

where $v_i$ is the $i$-th row of $V$ and the $< \cdot, \cdot >$ operator denotes inner-product.

**Definition 1.** *The signal content of $z$ with respect to matrix $W$, or simply the signal content of $z$, denoted by $z_S$ is defined as*

$$z_S = \arg\min_h \|h\|_2, \quad s.t., W \cdot (z - h) = 0. \tag{4}$$

*The null content of $z$ is also defined as $z_N = z - z_S$.*

**Lemma 1.** *We have*

$$z_S = \sum_{i=1}^{m} \alpha_i v_i^T \quad and \quad z_N = \sum_{i=m+1}^{n} \alpha_i v_i^T. \tag{5}$$

*Proof.* We write $h$ as the composition of orthonormal vectors $v_i$'s as $h = \sum_{i=1}^{n} \beta_i v_i^T$. We have

$$W(z - h) = \sum_{i=1}^{n} (\alpha_i - \beta_i) W v_i^T = \sum_{i=1}^{n} (\alpha_i - \beta_i) U S \underbrace{V v_i^T}_{q_i \in \mathbb{R}^n} \tag{6}$$

Since the rows of V are orthonormal, then $q_i = V v_i^T$ is a one-hot vector with its $i$-th element equal to 1. By substituting $q_i$ in (6) we obtain

$$W(z - h) = \sum_{i=1}^{n} (\alpha_i - \beta_i) U S_{[:,i]} = \sum_{i=1}^{m} (\alpha_i - \beta_i) U S_{[:,i]} = \sum_{i=1}^{m} s_i (\alpha_i - \beta_i) U_{[:,i]}, \tag{7}$$

where $S_{[:,i]}$ and $U_{[:,i]}$ are the $i$-th columns of $S$ and $U$, respectively. Note that, since $S$ is a diagonal matrix, we have $S_{[:,i]} = 0, \forall i \in \{m + 1, \cdots, n\}$, thus reducing the summation from $n$ to $m$ components. Also, for $i \in \{1, \cdots, m\}$, $S_{[:,i]}$ is a column vector with only one non-zero element, $s_i$, at the $i$-th dimension.

As a result, to obtain $W(z - h) = 0$, we must have $\beta_i = \alpha_i, \forall i \in \{1, \cdots, m\}$. Since $v_i$'s are orthonormal, we have $\|h\|_2 = \sqrt{\sum_{i=1}^{n} \beta_i^2}$. Hence, to minimize $\|h\|_2$, we set $\beta_i = 0, \forall i \in \{m + 1, \cdots, n\}$. Therefore, $z_S = \sum_{i=1}^{m} \alpha_i v_i^T$. The null content $z_N$ can be then computed as $z_N = z - z_S = \sum_{i=m+1}^{n} \alpha_i v_i^T$. $\square$

**Definition 2.** *The normalized signal and null contents of $z$ are defined as $C_S(z) = \frac{\|z_S\|_2^2}{\|z\|_2^2}$ and $C_N(z) = \frac{\|z_N\|_2^2}{\|z\|_2^2}$, respectively. We have $C_S(z) + C_N(z) = \frac{\sum_{i=1}^{m} \alpha_i^2}{\sum_{i=1}^{n} \alpha_i^2} + \frac{\sum_{i=m+1}^{n} \alpha_i^2}{\sum_{i=1}^{n} \alpha_i^2} = 1$.*

### 3.2 DEFENSE 1: OBFUSCATING NULL CONTENT OF FEATURE VECTOR

We propose to remove all the content in feature vector that is irrelevant to the public attribute. To do so, given a feature vector $z$, we find a minimum-norm vector $z'$ that generates the same prediction for the public attribute as $z$, i.e., $M_2(z') = M_2(z)$. Formally,

$$z' = \arg\min_h \|h\|_2, \quad s.t., M_2(h) = M_2(z). \tag{8}$$

| Removing null content | Removing part of signal content |
|---|---|
| **Training:** | **Training:** |
|   **Server** trains $M_2 \circ M_1$ with $(x_i, y_i^{\mathrm{pub}}), i \in [N]$ |   **Server** trains $M_2 \circ M_1$ using $(x_i, y_i^{\mathrm{pub}}), i \in [N]$ |
|   **Adversary** computes $z_i = M_1(x_i), i \in [N]$ |   **Server** augments $M_1 \rightarrow \widetilde{M_1}$ |
|   **Adversary** computes $z_i \rightarrow z_{i\mathcal{S}}, i \in [N]$ |   **Server** freezes $\widetilde{M_1}$ and fine-tunes $M_2 \circ \widetilde{M_1}$ |
|   **Adversary** trains $M_3$ with $(z_{i\mathcal{S}}, y_i^{\mathrm{pri}}), i \in [N]$ |   **Adversary** computes $\widetilde{z}_i = \widetilde{M_1}(x_i), i \in [N]$ |
| |   **Adversary** trains $M_3$ with $(\widetilde{z}_i, y_i^{\mathrm{pri}}), i \in [N]$ |
| **Inference:** | |
|   **Client** computes $z = M_1(x)$ | **Inference:** |
|   **Client** computes $z \rightarrow z_o$ |   **Client** computes $\widetilde{z} = \widetilde{M_1}(x)$ |
|   **Client** sends $z_o$ to server |   **Client** sends $\widetilde{z}$ to server |
|   **Server** computes $\hat{y}^{\mathrm{pub}} = M_2(z_o)$ |   **Server** computes $\hat{y}^{\mathrm{pub}} = M_2(\widetilde{z})$ |
|   **Adversary** computes $z_o \rightarrow z_{\mathcal{S}}$ |   **Adversary** computes $\hat{y}^{\mathrm{pri}} = M_3(\widetilde{z})$ |
|   **Adversary** computes $\hat{y}^{\mathrm{pri}} = M_3(z_{\mathcal{S}})$ | |

Figure 2: (left): Training and inference for defense 1 (client removes null content of $z$). (right): Training and inference for defense 2 (client removes low-energy content of $z$). $z_o$ is the feature vector in which the null content is obfuscated. $\widetilde{M_1}$ is constructed by augmenting $M_1$ with the module that removes the signal content.

Due to the complex (nonlinear) nature of deep networks, finding such a vector would require doing multiple backpropagations on $M_2$ for each given $z$. This is, however, not feasible for resource-constrained edge devices. To address this problem, we relax (8) such that the constraint holds for the first layer of $M_2$ (server-side model), i.e., we modify the constraint to $Wz' = Wz$, where $W$ is the weight matrix of the first layer of $M_2$. As discussed in Section 3.1, the solution to this relaxed optimization problem is $z_{\mathcal{S}}$, the signal content of $z$. Removing or obfuscating the null content of $z$ does not change the model prediction on public attribute. It, however, might harm the private accuracy since part of the null content of $z_{\mathcal{N}}$ might fall into the signal content of the first linear layer of $M_3$. The method is described in Figure 2 (left).

At inference time, to obfuscate $z_{\mathcal{N}}$, the client constructs $z_o$ using either of the following methods.

- Client constructs $\mu = \sum_{i=m+1}^{n} \eta_i v_i^T$, with coefficients, $\eta_i$, chosen at random, and sends $z_o = z + \mu$ to the server. The adversary can recover $z_{\mathcal{S}} = V_{1:m}^T \cdot V_{1:m} \cdot z_o$ but cannot recover $z_{\mathcal{N}}$.
- Client computes the signal content of $z$ and sends $z_o = z_{\mathcal{S}} = \sum_{i=1}^{m} \alpha_i v_i^T$ to the server.

For the first case, since $\mu$ is independent of $z$, the client can compute it offline, e.g., when the edge device is plugged in and not in use, and store it for later use. The second approach does not require storage on the edge device, but an extra computation, equal to the complexity of computing the first layer of $M_2$, has to be done during inference to extract $z_{\mathcal{S}}$. We next propose a method that reduces the extra cost to only a fraction of the computation cost of the first layer of $M_2$.

## 3.3 DEFENSE 2: DISCARDING LOW-ENERGY SIGNAL CONTENT OF FEATURE VECTOR

In the first defense method, we proposed to discard the content of the feature vector that will be removed by the first layer of $M_2$. The following layers of $M_2$ will further remove more content from feature vector. Hence, we can potentially discard more content from $z$ and still get similar prediction as the original feature vector. For a linear layer, following the same process in Section 3.1, the output is computed as:

$$W \cdot z = W \cdot z_{\mathcal{S}} = \sum_{i=1}^{m} \alpha_i U \cdot S \cdot q_i = \sum_{i=1}^{m} \alpha_i U \cdot S_{[:,i]} = \sum_{i=1}^{m} s_i \alpha_i U_{[:,i]}, \qquad (9)$$

where $s_i$ is the $i$-th eigenvalue in $S$, $\alpha_i$ is defined in (3), and $U_{[:,i]}$ denotes the $i$-th column of $U$. From (9) we observe that components with larger $s_i \alpha_i$ are contributing more to the layer output since $||U_{[:,i]}||_2 = 1$ for all columns of $U$. As such, we approximate $z \rightarrow \widetilde{z}$ by only keeping $m' < m$ components of the right-hand-side of (9) that have the largest coefficients. Unlike null content filtering, removing signal content will affect the public accuracy as it changes the main network output, but can further reduce the private accuracy. To improve public accuracy when removing signal content of features, the server fine-tunes $M_2$ on $\widetilde{z}$. The method is described in Figure 2 (right).

Since $s_i$ and $U_{[:,i]}$ are fixed at the inference time, the client only needs to send the selected $\alpha_i$ values along with their indices to the server; the server knows $s_i$ and $U$, and can reconstruct $\tilde{z}$ accordingly. The edge-computation cost of this process is $m'/m$ times the computation cost of the first layer of the server model. We do experiments in settings where $m'/m$ is about $1\%$. Hence, the computation cost of our method is only a small fraction of computing a single layer of the network. Moreover, since $m' \ll m$, our method also incurs a much smaller communication cost compared to sending the entire feature vector to the server.

## 4 EXPERIMENTS

### 4.1 SETUP

**Datasets.** We perform our experiments on four visual datasets listed below.

- **EMNIST** (Cohen et al. (2017)) is an extended version of the MNIST dataset where the labels are augmented with writer IDs. We selected 13000 samples from EMNIST written by 100 writers with 130 examples per writer. We then split this dataset into 10000, 1500, and 1500 training, validation, and test sets. We use the digit label and writer ID as the public and private attributes, respectively.

- **FaceScrub** (Ng & Winkler (2014); FaceScrub (2020)) is a dataset of celebrity faces labeled with gender and identity. We use gender and identity as the public and private attributes, respectively. In experiments, we cropped the face region using the bounding boxes in the image annotations and resized images to $50 \times 50$.

- **UTKFace** (Zhang et al. (2017)) is a dataset of face images labeled with gender and race, which we use as the public and private attributes, respectively. We cropped the face region using the bounding boxes in the image annotations and resized images to $50 \times 50$.

- **CelebA** (Liu et al. (2015)) is a dataset of celebrity images. Each image is labeled with 40 binary attributes. Out of these, we select "Smiling" as the public attribute and {Male, Heavy_Makeup, High_Cheekbones, Mouth_Slightly_Open, Wearing_Lipstick, Attractive} as private attributes. These attributes have near balanced distribution of positive and negative examples. In experiments, we cropped the face region using the bounding boxes in the image annotations and resized images to $73 \times 60$.

**Model architecture.** We present the experimental results on a model used in prior work (Song & Shmatikov (2019)). The model architecture and baseline test accuracy results are summarized in Table 1 and Table 2 in Appendix.

**Adversary capabilities.** We use the same architecture for adversary's model $M_3$ as the server model $M_2$. The model $M_3$ is trained using the features extracted by $M_1$ and the associated private labels. We also assume that the adversary knows the parameters of $M_1$ and $M_2$.

**Training settings.** We use Adam optimizer with an initial learning rate of $0.001$ and drop the learning rate by a factor of 10 after 20 and 40 epochs. All models including the adversary and main models are trained for 50 epochs unless stated otherwise.

### 4.2 EVALUATIONS

We start our analysis by computing the null and signal contents in every layer of $M = M_2 \circ M_1$. Figure 3 (left) shows the content of the input remained at each layer; for the $i$-th layer, this content is computed as $\prod_{j=1}^{i} C_{\mathcal{S}}(z_j)$ where $z_j$ denotes the activation vector at the $j$-th layer and $C_{\mathcal{S}}(\cdot)$ is defined in Section 3.1. As the feature vector propagates through network layers, more content is gradually removed from $z$ until the model outputs the prediction on the public task. We also split the network at different layers and measure the private accuracy of $M_3$ trained with the feature vector. As seen in Figure 3 (right), the private accuracy also decreases as we get closer to the output layer, indicating that the discarded content contained information relevant to the private attribute.

To reduce the privacy leakage, we proposed to filter out the null content of the feature vector. Figure 4 shows the private accuracy for different split layers. Removing the null content reduces the private

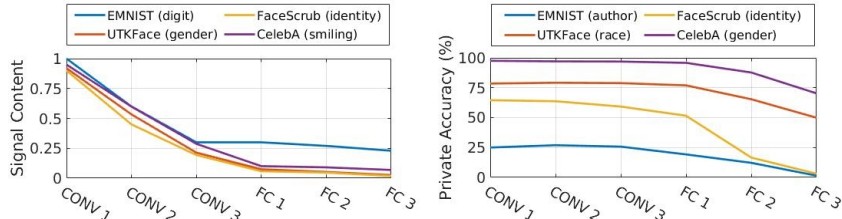

Figure 3: (left) Remaining signal content at the output of each layer of the main model, (right) private accuracy for different split layers. The content of the input significantly decreases in deeper layer. Similarly, the private accuracy decreases, indicating that the discarded content contained information relevant to the private attribute.

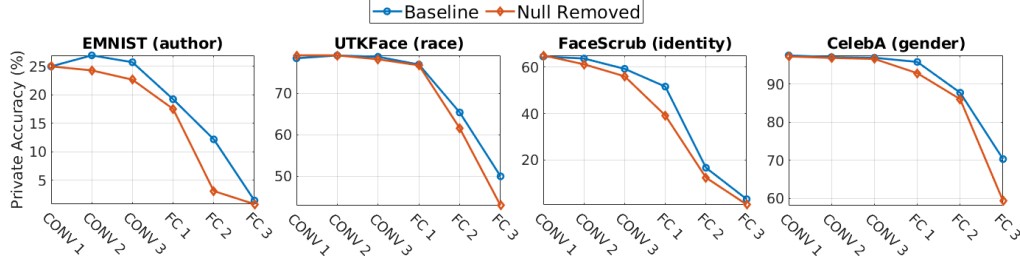

Figure 4: The effect of removing the null content on the private accuracy with different split layers.

accuracy without affecting the public accuracy. Moreover, splitting the network in deeper layers improves the privacy. To further reduce the private accuracy, we discard the low-energy components of the signal content and only keep $m'$ features. Figure 5 illustrates the effect of $m'$ on the public and private accuracy, when the network is split at the CONV-3 layer. As seen, by setting $m'$ to a small value, our method achieves high privacy (low private accuracy) at the cost of a small reduction in public accuracy. In general, the privacy can be controlled using two factors:

- **The split layer:** As we go deeper in the network, i.e., when the edge device performs more computation, better tradeoffs can be achieved. To show this effect, we perform signal-content removal at different layers, with $m'$ set such that the public accuracy is reduced by 1%. The corresponding private accuracy is shown in Figure 6.

- **Number of signal components sent to the server:** For the same edge-computation (a given split layer), the number of preserved features ($m'$) can be tuned so as to achieve a desired tradeoff between utility (higher public accuracy) and privacy (lower private accuracy). Figure 5 shows the results for the setting that the network is split at the input of the CONV-3 layer.

**Comparison to Pruning.** Similar to our approach, pruning network layers can eliminate features that do not contribute to the public attributed. In the following, we compare our method with pruning in terms of public and private accuracy. We split the network from the middle layer, i.e., at the input of the $FC$-1 layer. For our method, we keep the top $m'$ components of $z$ from its signal content and filter out the rest. For pruning, we keep $m'$ elements in $z$ and set the rest to zero. We adopt the pruning algorithm proposed by (Li et al. (2016)) which works based on the $L_1$ norm of the columns of the following layer's weight matrix. After pruning, we fine-tune $M_2$ to improve the public accuracy. Figure 7 shows the public and private accuracy for each dataset. As seen, with small $m'$, both our method and pruning achieve a low private accuracy. Pruning, however, significantly reduces the public accuracy as well. For example, for the UTKFace dataset, with $m' = 1$, both methods result in a private

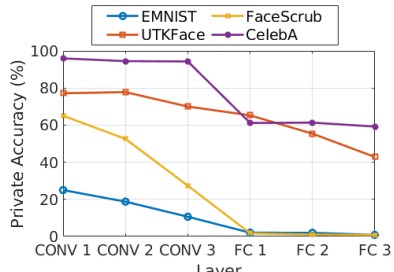

Figure 6: Privacy accuracy versus split layer. The number of preserved components of signal content is set such that the public accuracy is reduced by 1%.

accuracy close to the random guess. However, pruning reduces the public accuracy from 92.25% to 53% (also close to random guess), whereas our method keeps the public accuracy at 88.63%.

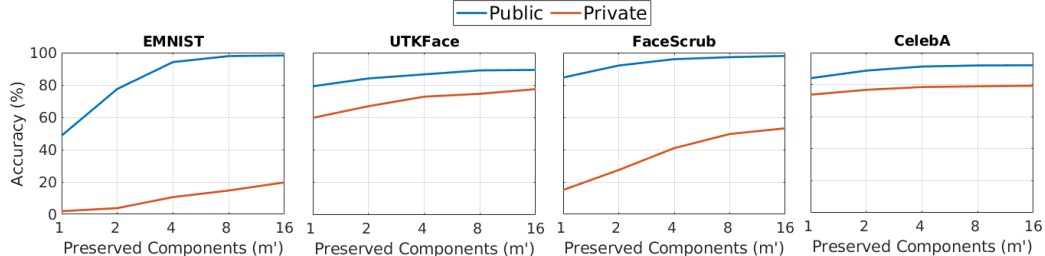

Figure 5: The effect of the number of preserved features ($m'$) on the public and private accuracy when the network is split at the input of *CONV-3* layer.

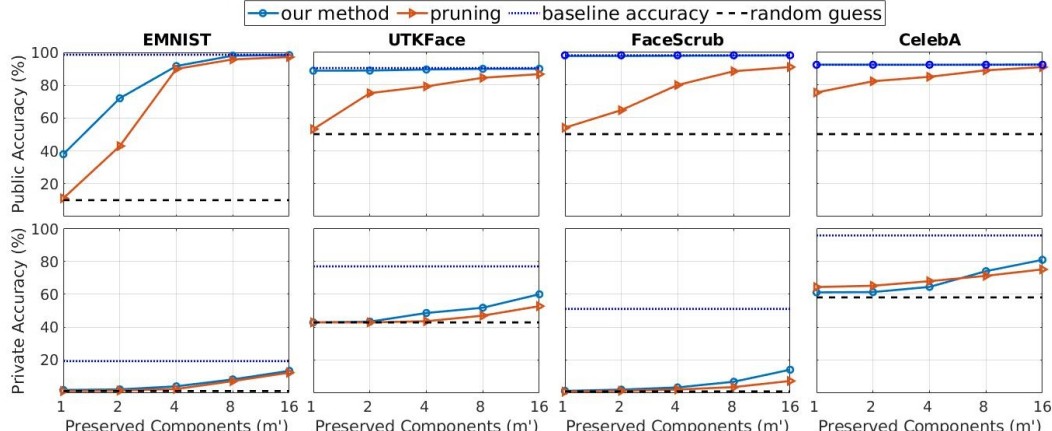

Figure 7: Comparison between our method and feature pruning. Both methods reduce the private accuracy. Pruning, however, significantly reduces the public accuracy.

**Comparison with adversarial training.** We implement adversarial training framework proposed by (Feutry et al. (2018)) and present the utility-privacy tradeoff in Figure 8. To achieve the best tradeoff for adversarial training, we train the models in multiple settings with different $\gamma$ parameters (Eq. 2) in range of $[0.1, 1]$. Note that, unlike our method, adversarial training assumes that the private attribute is known at training time. Despite this, Figure 8 shows that our method achieves a better utility-accuracy tradeoff than adversarial training.

We also do experiments for the case with multiple (unseen) private labels. Specifically, we consider the CelebA model trained to detect "smiling" and evaluate two methods, 1) our method: we keep only $m' = 1$ component from the signal content of feature vector and then train one adversary model per private attribute, and 2) adversarial training: we first adversarially train an $M_1$ model to obfuscate "gender," and then train one model for each private attribute.

For both of the above methods, the network is split at the input of the $FC$-1 layer. Figure 9 shows the results. Our method outperforms adversarial training method on both public and private accuracy. In our method, the accuracy on all private attributes are significantly lower than the baseline private accuracy. The only exceptions are "high cheekbones" and "mouth open" attributes, which have correlations with public attribute, that is, a smiling person likely has high cheekbones and their mouth open. The correlation between public and private attributes causes the signal content of server and adversary's models to have large overlaps and, hence, results in high private accuracy. The adversarially trained model successfully hides the information that it has been trained to obfuscate (the "gender" attribute). Such a model, however, fails to remove information of other attributes such as "makeup" or "lipstick". This highlights the applicability of our method in practical setting as a generic obfuscator compared to specialized techniques such as adversarial training.

**Ablation study on CONV and FC layers.** We compare the performance of our method on CONV and FC layers. To do so, we train two networks on the UTKFace task, (1) a network with 10 CONV

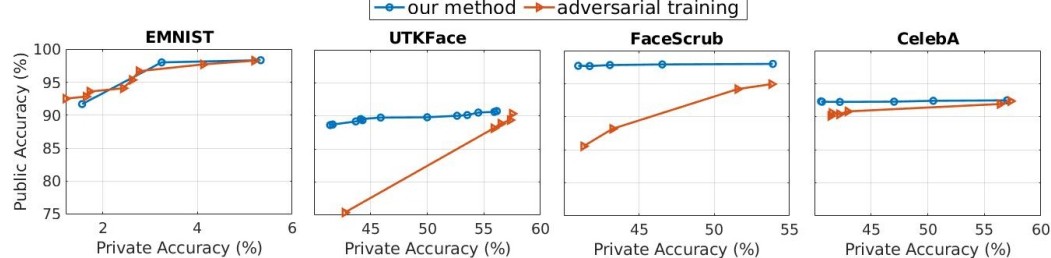

Figure 8: Comparison between our method and adversarial training. For a given privacy level, our method provides higher utility (higher public accuracy) compared to adversarial training method.

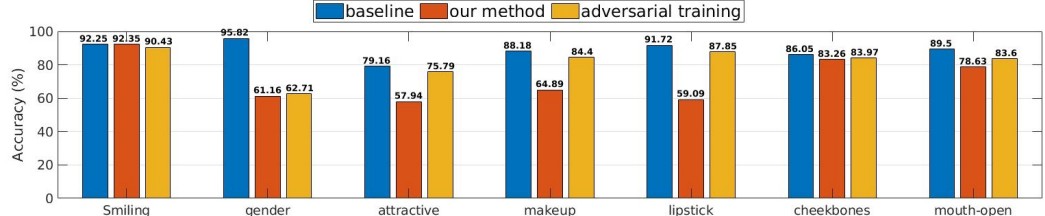

Figure 9: Accuracy on public ("smiling") and private attributes. Our method obfuscates the feature vector without the knowledge of private attribute at training or inference times. Adversarial training method maximizes the accuracy on "smiling," while minimizing the accuracy on "gender." As can be seen, our method reduces accuracy on all private attributes. The adversarially trained model successfully reduces accuracy on "gender" attribute, but fails to remove information of other attributes. This highlights the applicability of our method in practical settings as a generic obfuscator compared to specialized techniques such as adversarial training.

layers each with 16 output channels, and (2) a network with 10 FC layers each with 2304 neurons. Both networks have an extra FC layer at the end for classification. The number of channels/neurons are chosen such that the total number of output features at each layer is the same for the two networks. The public accuracy of the 10-CONV and 10-FC networks is $89.26\%$ and $89.07\%$, respectively. Figure 10 shows the public and private accuracy when we remove low-energy components of the signal space at different layers. The number of preserved features, $m'$, at each layer is chosen such that the public accuracy is maintained. As seen, the 10-FC network achieves a lower private accuracy compared to the 10-CONV network. The reason is that CONV layers are known to be generic feature extractors, while FC layers are more specialized toward the public attribute.

## 5 CONCLUSION

We proposed a private inference framework, in which edge devices run several layers locally and obfuscate the intermediate feature vector before sending it to the server to execute the rest of the model. For obfuscation, we proposed to remove information that is not relevant to the main task or does not significantly change the predictions. Specifically, we developed two methods of removing the content of the feature vector in the null space of the following linear layer and also removing the low-energy content of the remaining signal. We showed that, unlike existing methods, our methods improve privacy without requiring the knowledge of private attributes at training or inference times.

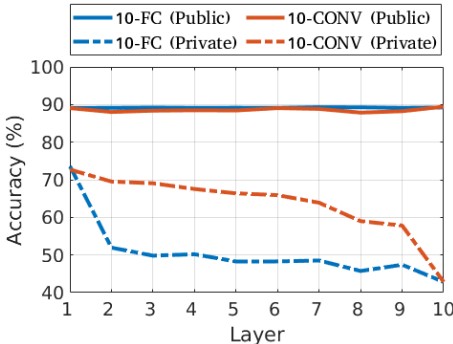

Figure 10: Comparing the performance of our method on the 10-FC and 10-CONV networks. The number of preserved features at each layer is set such that the public accuracy is maintained.

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

## A  APPENDIX

Table 1: Network Architecture. Each row shows a split layer, e.g., at layer 1, the raw data and, at layer 6, the input to the last fully-connected layer is sent to the server.

| Split Layer | | | | |
|---|---|---|---|---|
| 1 | CONV $(3, 16)$ | ReLU | Max-Pooling $(2 \times 2)$ | Batch-Normalization |
| 2 | CONV$(3, 32)$ | ReLU | Max-Pooling $(2 \times 2)$ | Batch-Normalization |
| 3 | CONV $(3, 64)$ | ReLU | Max-Pooling $(2 \times 2)$ | Batch-Normalization |
| 4 | FC(128) | ReLU | - | Batch-Normalization |
| 5 | FC(64) | ReLU | - | Batch-Normalization |
| 6 | FC(n_classes) | Softmax | - | - |

Table 2: Model accuracy for public and private attributes of different datasets.

| Dataset | MNIST | UTKFace | FaceScrub | CelebA | | | | | |
|---|---|---|---|---|---|---|---|---|---|
| number of classes | 10 | 2 | 2 | 2 | | | | | |
| public attribute | digit | gender | gender | smiling | | | | | |
| public accuracy (%) | 98.60 | 90.25 | 97.90 | 92.25 | | | | | |
| number of classes | 100 | 5 | 530 | 2 | 2 | 2 | 2 | 2 | 2 |
| private attribute | writer | race | identity | gender | makeup | cheekbones | mouth-open | lipstick | attractive |
| private accuracy (%) | 26.93 | 79.18 | 65.52 | 97.53 | 90.00 | 86.29 | 92.94 | 93.40 | 80.46 |

