# OpenReview forum: "Private Split Inference of Deep Networks"
_ICLR.cc/2021/Conference — Reject_

### Official Review · AnonReviewer2 · 2020-10-26
**Interesting idea but needs better positioning**

**Rating:** 5
**Confidence:** 3

**Review:**

This paper proposes model splitting as a method to perform private inference from an edge device to a cloud provider. The idea being that an edge device owns layers 0 -> i of a model, M, and the cloud device owns layers i+1 -> N. Then, when an edge device wants to perform inference on a private input, x, they send M_i(x) instead of x to the cloud device, who then completes the prediction task. The claim is that if M_i(x) is learned representation that only retains features that are predictive of the public label, there should be no private information leaked about the input to an eavesdropping adversary who can also observe M_i(x). The authors achieve this by finding representations that are approximately equivalent in the following layer (i+1), but have much smaller norms than the original feature vectors. At inference time, the client either directly constructs the signal content part of the feature or randomises over the nuisance components. Experiments on four datasets point to the effectiveness of this idea, and that simple baselines such as adversarial training or pruning perform worse. I thought the core experiments were sound, but I am not totally convinced by the motivation of the approach.

1. I don't understand why cryptographic solutions are not viable for this kind of problem? If I understand correctly, the server is trusted, the only assumed vulnerable step is the transmission of data between the edge device and cloud provider. If this is correct, then would a simple PKE scheme really be prohibitively expensive? Even a homomorphic set-up seems somewhat tolerable (depending on your threat model) given the experiment datasets used in this work. For example, message sizes of CryptoNet for EMNIST size data is ~91KB [1].


2. Following on from (1), I found it hard to reason about trade-offs between public accuracy, private accuracy, computational expense of the defense, bandwidth overheads etc. For example, if I sacrifice 1% of public accuracy for 5% private accuracy, is that a good trade-off? Who makes that decision? The edge device or the cloud provider? The paper would have been much improved if these kinds of questions were captured by a formal threat model.


3. Most the experiments concentrated on public / privacy accuracy trade-offs on problems with a balanced distributions over private attributes. How does this scheme cope with more realistic problems where the distributions are highly skewed?


4. How weak is the approximation to eq. (8)? I couldn't find any ablation studies to justify that this approximation is reasonable. Additionally, what are the computational overheads of extracting z_S?


5. In Figure 4, why is there a large drop at the final Conv layer?


Minor: '..do not contribute to the public attributed." -> "..do not contribute to the public accuracy."?


[1] Gilad-Bachrach, Ran, et al. "Cryptonets: Applying neural networks to encrypted data with high throughput and accuracy." International Conference on Machine Learning. 2016.

---

> ### Author Response · Authors · 2020-11-19
> **Response to Reviewer 2**
>
> Thank you for your comments. We believe most of the comments can be addressed in the paper by improving the threat model section of the paper. We will use the extra 1-page at the revision phase to refine the threat model. In the following, we provide our responses to the comments.
> 1. About using cryptographic solutions:
> * We assume that the server is honest-but-curious, i.e., it performs the inference on the public attribute but will potentially try to extract private information as well. As a result, the server is not trusted and, in fact, is the adversary itself. Therefore, the PKE scheme is not a viable solution.
> * We envision a scenario where the server is providing a service to millions of users (e.g., in cases of Amazon Alexa or Apple SIRI), and users expect low communication and fast response. Cryptographic solutions such as Homomorphic Encryption (HE) and Garbled Circuits (GC) provide provable security but are not applicable for large-scale adoption for several reasons, including: (1) HE and GC increase the communication cost and delay by orders of magnitude. This might be acceptable for applications such as healthcare where a few minutes of delay is tolerable but is not applicable in our scenario, and (2) lack of efficient support for nonlinear operations. Specifically, HE does not support nonlinear operations such as ReLU, which is an important component in neural networks. In CryptoNets, ReLUs were replaced with square function which significantly reduces the accuracy. In GC, nonlinear operations are supported but at the cost of large communication overhead, which makes it not applicable for our use-case.
> 2. About the threat model: Upon providing the service, the server also provides a profile of the utility, privacy, and edge-compute to the client. The client then decides on the best tradeoff according to the computational resources and also the desired level of privacy. Such mechanisms are already in use in ML-on-the-edge applications. For example, in the application of unlocking the phone by face recognition, the client can specify the required precision in face recognition, where a lower precision will provide higher utility at the cost of lower security. We will make the threat model clearer in the revision.
> 3. About distribution of private attributes: Our method does not require the distribution of private features to be balanced. In fact, our method does not use the private attributes at all and, hence, does not make any assumptions on them. In experiments, we showed the results on balanced attributes so as to make the public and private accuracy results more comparable and interpretable.
> 4. About the approximation to eq. (8):
> * Finding the exact solution to (8) requires multiple steps of backpropagation on M2 for each input. This will make the method computationally expensive and defeats the purpose of splitting the network for efficient inference. As a result, we used an approximation based on the first layer of the server model, which is computationally efficient and provides good empirical results.
> * The computational cost of our method is equivalent to m’/m times the computation cost of the first layer of the server model. In experiments, we worked in settings where m’/m is about 1%. Hence, the computation cost of our method is only a small fraction of computing a single layer of the network. Moreover, since m’<<m, we also achieve a much smaller communication overhead compared to sending the entire activations to the server. We will make this clearer in the revised paper.
> 5. About Figure 4: Please note that the m’ required to maintain the public accuracy is different for different layers since the data dimensionality varies across layers (the spatial size of activations is smaller in deeper layers). In Figure 4, each curve shows the results for the same m’ for all layers, which causes the misperception that the last convolutional layer has a drop in accuracy. We will edit the figure and make this point clearer in the revision.

---

### Official Review · AnonReviewer1 · 2020-10-28
**feasibility of the proposed method in deep neural networks**

**Rating:** 5
**Confidence:** 5

**Review:**

This paper tackles a timely problem of privacy leakage on the edge devices when applying deep neural networks. Instead of mitigating the leakage of a set of private attributes, the proposed method tries to remove the information irrelevant to the primary task. The proposed method does not need to identify the private attributes. The main contribution of this paper is the two proposed approaches for removing “null content” and “signal content.” The evaluations of the proposed approach are conducted on four image datasets.

Pros:
1. The idea of removing irrelevant information instead of private attributes is an interesting idea.
2. The paper is well organized and well written.
3. The experimental evaluation is comprehensive. Feature pruning and adversarial training are included in the evaluation.

Cons:
1. The key concern about the paper is the feasibility of the proposed methods in deep neural networks. Both proposed feature-removing methods are derived from a single linear layer. However, in many cases and even shown in the evaluation, the device side may process more than one layer of neural networks. In addition, the convolution layer is often deployed as the first layer in neural networks. It would be great if the proposed methods can be extended to multiple layers and multiple types of neural networks.
2. The adversary uses the same architecture in the paper. However, the adversary can choose to use a more complex model to extract the privacy attributes in the evaluation. The different architecture may cause the failure of the proposed methods. It would be nice if more adversarial models can be evaluated in the paper.
3. In Figure 4, the proposed methods do not perform well in balancing the utilities and privacy achieved. It is hard to tell if the better tradeoffs are due to the deeper layers or fully connected layers. From Figure 4, it seems the proposed methods do not perform well on the convolutional layers.
4. The experiments only evaluate on a six-layer neural network, which is not a “deep network” claimed in the title. It would be great if the paper can evaluate the performance on other architectures and deeper models.
5. The algorithms in Figure 2 are hard to understand.


Minor comments:
1. In Equation 2 it should be “M2 * M1” instead of “M1 * M2”
2. Page 6 Figure 4 shows that the information leakage can be controlled using the following factors “factors”

---

> ### Author Response · Authors · 2020-11-19
> **Response to Reviewer 1**
>
> Thank you for your comments. We provide our responses in the following.
>
> 1. About feasibility: In our split learning framework, we assume that the deep neural network is partitioned into two parts, M1 and M2, where M1 and M2 can have any number of layers. Our defense methods remove the null space content of the output of M1 (and the low-energy portion of its signal space) before sending it to the server, where the signal and null space are defined according to the first layer of M2. Our methods work for both linear and convolutional layers and with networks with any number of layers on the device- or server-side. The client’s capability of choosing the split layer is, in fact, one of the unique features of our method compared to the existing techniques and provides a tradeoff between the privacy and computation on the edge. We will make this clearer in the paper.
>
> 2. About adversary’s architecture: It is true that the adversary’s architecture can affect its performance. We, however, used the same architecture for evaluating the performance of our methods and the baseline techniques, which provides a fair experimental setup. The architecture used is also the same as the one used in recent related work. Nevertheless, we will include an ablation study on adversary’s architecture in the revised version of the paper.
>
> 3. About performance of the method on balancing the utility and privacy:
> * Figure 4 shows the performance of our method when we preserve only a few high-energy features from the signal space. As mentioned in the paper, our method achieves better (lower) private accuracy for deeper layers. We also observed that the required number of preserved features, m’, to maintain the public accuracy is generally larger for convolutional layers compared to FC layers. Please, however, note that the notion of feature is different for convolutional and FC layers. Figure 4 causes the misperception by showing the results for the same m’ across both convolutional and FC layers. We will edit the figure and make this point clearer in the revision.
> * Nevertheless, it is true that our method achieves better results for FC layers compared to convolutional layers. The reason is that convolutional layers are by nature generic feature extractors, while FC layers are more specialized to the public attribute. To see this, we did experiments on the UTKFace dataset with two networks: 1) a network with 10 convolutional layers each with 16 output channels, and 2) a network with 10 FC layers each with 2304 neurons. The number of channels/neurons are chosen such that the total number of output features at each layer is the same for the two networks. At each layer, we remove the signal content to a degree that the public accuracy is preserved and, then, compute the private accuracy. The result is provided in the following anonymous link:
> [link to figure, please click on PublicPrivate.png](http://s000.tinyupload.com/index.php?file_id=67859634137858253341).
> As seen, the public accuracy remains high for both networks, while the private accuracy decreases for deeper layers with the FC network achieving better (lower) private accuracy results. We will add this evaluation to the revised paper.
> * Overall, we showed in the paper that our methods achieve better utility-privacy tradeoff compared to existing techniques. As an example, in Figure 7, we compared the performance of our method with a recently proposed adversarial training (AT) framework. In AT, the model is trained to obfuscate a known private attribute. We showed that, when evaluating on the private attribute that AT has been trained on, our method matches the private accuracy of AT despite not having access to the private attribute. Moreover, when evaluating on other unknown private attributes, our method significantly outperforms AT. As mentioned in the paper, this highlights the applicability of our method in practical settings as a generic obfuscator compared to specialized techniques such as AT.
>
> 4. About using deeper networks in experiments: To conduct fair experiments, we used the same architectures used in related work. Our method can certainly be applied to deeper networks. Please note that a deeper client-side model allows the edge device to compress the activations more due to the higher capacity of the model. As a result, the device can discard more private information from the feature vector and, hence, obtain a better utility-privacy tradeoff.
>
> 5. We will describe the algorithm in Figure 2 more clearly in the revision.

---

### Official Review · AnonReviewer3 · 2020-10-28
**This paper proposes a matrix decomposition method to model the public information and the private information. It tries to compare to noise injection, information bottleneck and adversarial training methods to obfuscate the private information. Extensive experiments show the effectiveness of the proposed approach.**

**Rating:** 5
**Confidence:** 4

**Review:**

The paper is well organized with sufficient background discussion and the related works. The annotation and methodology introduction is clear and mostly without error. This paper lies in an interesting setting of client-server, by sending the shared representation z while decoding the public and private feature at the server side.

The private information is not necessarily to be orthogonal to the public information. Thus, by the proposed method, without the semantic labels of the private attribute, it is not clear why the orthogonal to public feature would necessarily to be private feature. Yet, public feature would contain private feature, and private feature would contain public feature.

From Figure 1, the authors actually claims to utilize adversary training to predict the secrete attribute, which mostly will utilize the private labels. While in the conclusion, the authors mentioned they do not require the knowledge of private attributes, which is a contradictory. Meanwhile, there is no discussion relating to M3 in Section 3 “proposed method”.

The datasets in the experiment are mostly towards face related data. It would be more convincing with a more general purpose dataset, such as DomainNet, which is aiming at object categories from different resources, sketch, photos, arts, where public information could be the categories, and the private information is the resource type.

From the experimental settings, those public information and private information, by the datasets’ setting, are naturally independent, i.e. gender and race in UTKFace, these two are naturally orthogonal. If the public and private information are with some correlation, would the proposed framework still work under this situation?

---

> ### Author Response · Authors · 2020-11-19
> **Response to Reviewer 3**
>
> Thank you for your comments. We provide our responses in the following.
>
> * About orthogonality of public and private information: As you mentioned, the public and private information could be indeed highly correlated. Our proposed methods do not assume that the private information is orthogonal to the public information. We provided an example in Figure 7 (Figure 9 in the revised manuscript), where the private labels “high-cheekbones” and “mouth-slightly-open” are correlated with the public label “smiling” and, thus, their private accuracy does not reduce much when the defense is used. The information leakage to private attributes is inevitable in such cases. Our method, however, exploits the fact that the part of the private information that lies in the null-space of the server model is not necessary for inference of the public label and, hence, can be discarded. Referring to Figure 7 again (Figure 9 in the revised manuscript), the private attributes of “gender”, “attractive”, “makeup”, and “lipstick” are not correlated with the public attribute and, as a result, their private accuracy drops close to random guess after applying our method.
>
> * About having the knowledge of private attributes: Figure 1 presents the threat model, not our privacy-preserving method. Our methods do not use private attributes or adversarial training to provide privacy and only need public labels to train the model M=M2 o M1. We simulate the adversary via M3 in Figure 1 as an aftermath process only to evaluate the privacy of our model. We will provide more details on our threat model at the beginning of Section 2.
>
> * About datasets: Thank you for your suggestions about the new datasets. In our paper, we focused on classification models and followed the recent related work on ML privacy for dataset selection. The DomainNet dataset you suggested is indeed very interesting for privacy analysis. We will consider it for our future work. Please also note that the EMNIST dataset we used is similar in that it has different categories (digits) each with a private writer ID.
>
> * About experimenting with correlated public and private information: This setting is indeed very important. We provided evaluation of such correlated public and private attributes in Figure 7 (Figure 9 in the revised manuscript). Please refer to our response in the first bullet point for more details.

---

### Author Response · Authors · 2020-11-24
**Paper is revised**

We would like to thank the reviewers for their valuable feedback. We have updated the paper to address the comments.

The major changes made to the paper are as follows:
* We added more details on our threat model, described the capabilities of the clients and the server, and explained our target use-case.

* We revised figure 4 and split it into thee figures 4-6 for more clarity. The figures evaluate different aspects of the tradeoffs between accuracy, privacy, and efficiency, described as follows:
  * The new figure 4 shows the effect of removing the null content on the private accuracy with different split layers.
  * The new figure 5 shows the effect of the number of preserved features on the public and private accuracy when the network is split at the input of CONV-3 layer.
  * The new figure 6 shows private accuracy versus split layer, where the number of preserved features is set such that the public accuracy is reduced by 1%.

* We added an ablation study on the performance of our method on CONV and FC layers. We showed that our method improves privacy when the network is split from either layer type. However, FC layers provide better accuracy at the same accuracy.

We believe the suggestions helped us to improve our submission. We hope that the reviewers have the chance to read our rebuttal and the updated paper and revise their scores.

---

### Decision · Program_Chairs · 2021-01-07
**Final Decision**

**Decision:**

Reject

**Comment:**

While reviewers believe that the motivation of the paper is strong and the idea is interesting the ultimate execution of the paper is not up to the standards of ICLR. I believe the biggest concern is the precise privacy guarantee of the method. As pointed out, it is an extremely strong assumption that the model structure of the adversary is known (or even approximately known). Standard privacy guarantees are either information theoretic or based in computational hardness. This work does not provide such guarantees. While there has been recent work on using adversarial learning to learn models that are robust to such adversaries, they have been heavily criticized within privacy and security communities due to the lack of such guarantees. I was not convinced by the authors response to such questions: there are plenty of cryptographic/privacy-preserving schemes that work in the honest-but-curious setting, and techniques that use the standard guarantee of differential privacy do not suffer from large slow downs.

Thus, I would urge the authors to modify this work so that it can leverage the guarantees of well-known cryptographic/privacy-preserving schemes. If done so, these arguments about privacy will go away and the paper will have a much better shot at acceptance.